

# Dynamic Thinning and Grounding Line Retreat in Porpoise Bay, Wilkes Land, East Antarctica

Matilda Weatherley[1], Chris R. Stokes[1], Stewart S. R. Jamieson[1], Sindhu Ramanath[2], Alessandro Silvano[3]

[1]Department of Geography, Durham University, Durham, DH1 3LE, UK
[2]Remote Sensing Technology Institute, German Aerospace Centre, 82234 Weßling, Germany
[3]School of Ocean and Earth Science, University of Southampton, Southampton, SO14 3ZH, UK

*Correspondence to*: Matilda Weatherley (matilda.weatherley@durham.ac.uk)

**Abstract.** The East Antarctic Ice Sheet (EAIS) is often considered less vulnerable to climate change than the West Antarctic or Greenland ice sheets, but some regions of the EAIS have been losing mass over recent decades. In particular, mass loss in
Wilkes Land, which overlies the Aurora Subglacial Basin, is thought to have accelerated over the past two decades. However, whilst several large outlet glaciers drain this region, few have been studied in detail. Here, we present new data on the recent ice dynamics of four outlet glaciers that drain into Porpoise Bay, Wilkes Land, which includes Holmes East, Holmes West, and Frost glaciers. We use optical satellite imagery, differential synthetic aperture radar interferometry, and a range of previously published datasets to describe changes in the ice-shelf front, grounding line position, ice surface velocity and ice
surface elevation over the last three decades. Our results reveal evidence of dynamic changes in the region, characterised by thinning of grounded ice and grounding line retreat, albeit with large uncertainties. We find an indication of Circumpolar Deep Water proximal to the continental shelf break that could access the glaciers through deep cross-shelf troughs, which is consistent with previous estimates of high rates of basal melting beneath their floating tongues/ice shelves. Our results also support previous observations of near-synchronous ice-shelf calving across Porpoise Bay's ice shelves, following the break-
out of multi-year sea ice, and find an additional recent calving event, further highlighting the vulnerability of this region to ongoing and future changes in ocean and sea-ice conditions.

## 1 Introduction

The Antarctic Ice Sheet is the largest ice sheet worldwide, containing 57.9 m of global mean sea level equivalent (SLE) (Morlighem et al., 2020). The Antarctic Ice Sheet comprises three regions: the East Antarctic Ice Sheet (EAIS), the West
Antarctic Ice Sheet (WAIS), and the Antarctic Peninsula. Recent work has mostly focussed on changes in the Antarctic Peninsula (van den Broeke, 2005; Berthier et al., 2012; Cook et al., 2014, 2016; Leeson et al., 2020) and pronounced mass loss in the WAIS (Pritchard et al., 2009; Mouginot et al., 2014; Rignot et al., 2014; McMillan et al., 2015; Feldmann and Levermann, 2015; Pattyn and Morlighem, 2020), which contains 5.3 m SLE (Morlighem et al., 2020). Much less work has focused on the stability of the EAIS, containing 52.2 m SLE, an order of magnitude larger than the WAIS (Morlighem et al.,
2020; Stokes et al., 2022).



Recent mass balance studies show accelerating mass loss from the Antarctic Ice Sheet over recent decades; the most recent assessment estimates 2671 ± 530 Gt of ice lost between 1992 and 2020, equating to 7.4 ± 1.5 mm of sea-level rise (Otosaka et al., 2023). This mass loss is dominated by loss from the WAIS (82 ± 9 Gt yr$^{-1}$) and the Antarctic Peninsula (13 ± 5 Gt yr$^{-1}$), with the EAIS remaining close to balance (3 ± 25 Gt yr$^{-1}$), albeit with larger uncertainties (Otosaka et al., 2023).

Mass loss from the WAIS accelerated from −37 ± 19 Gt yr$^{-1}$ (1992–1996) to −131 ± 21 Gt yr$^{-1}$ (2012–2016) then slowed to −94 ± 25 Gt yr$^{-1}$ (2017–2020) (Otosaka et al., 2023). Mass loss from the WAIS is concentrated in the Amundsen Sea Embayment, primarily from Pine Island and Thwaites glaciers (Mouginot et al., 2014; Gardner et al., 2018; Shepherd et al., 2019). Ice sheet mass is lost through ice flow acceleration, thinning, and grounding line retreat, which is predominantly driven by ice-shelf debuttressing due to warm ocean water melting the base of ice shelves (Mouginot et al., 2014; Rignot et al., 2014;

Fürst et al., 2016; Pattyn and Morlighem, 2020).

Comparatively, the EAIS has often been considered to be much less vulnerable to recent climate change, with the IMBIE mass balance estimates (e.g. Otosaka et al., 2023) revealing it is largely in balance or slightly increasing in mass (albeit with greater uncertainties than the WAIS). It has been noted that mass gain is generally concentrated in Dronning Maud Land and mass loss in Wilkes Land, with other areas close to balance (Gardner et al., 2018; Smith et al., 2020a; Otosaka et al., 2023).

However, there is emerging evidence of a longer-term trend of regional mass loss from Wilkes Land (Rignot et al., 2019; Smith et al., 2020a; Wang et al., 2021; Stokes et al., 2022), referred to as the 'weak underbelly' of the EAIS (Miles et al., 2016; Pelle et al., 2020). Wilkes Land is drained by 39 outlet glaciers and has received growing attention in the past three decades due to the emerging signal of mass loss (Gardner et al., 2018; Rignot et al., 2019; Schröder et al., 2019a; Shepherd et al., 2019; Smith et al., 2020a; Nilsson et al., 2022; Stokes et al., 2022, 2025). Mass loss has been linked to the intrusion of warm modified

Circumpolar Deep Water (mCDW) in sub-ice cavities through deep cross-shelf troughs, driving melt rates that are comparable to the Amundsen Sea Embayment (Rintoul et al., 2016; Silvano et al., 2016, Miles et al., 2016; Rignot et al., 2019). Indeed, observations have confirmed mCDW proximal to three key glaciers in Wilkes Land: Vanderford, Totten, and Moscow University glaciers (Greenbaum et al., 2015; Rintoul et al., 2016; Silvano et al., 2017; Picton et al., 2023). A recent study by Rignot et al. (2019) estimated the cumulative mass loss from each drainage basin in East Antarctica between 1979 and 2017,

with Totten (236 Gt) and Denman (191 Gt) glaciers contributing the greatest, followed by Frost (159 Gt) and Holmes (152 Gt) glaciers that both feed Porpoise Bay in the Wilkes Land. However, despite Frost and Holmes glaciers losing mass at potentially high rates, there have been few studies of these major outlet glaciers. Furthermore, Wilkes Land is underlaid by a bed that lies below sea level and deepens inland to the Aurora Subglacial Basin, containing several deep troughs (Morlighem, 2022), making it potentially vulnerable to marine ice sheet instability (MISI), which may have occurred during past warm periods

(Cook et al., 2013).

## 1.1 Porpoise Bay

Porpoise Bay (76° S, 128°E) is a 150 km wide bay that lies east of Moscow University Glacier (~300 km) and Totten Glacier (~550 km) (Fig. 1a). It overlies the eastern Aurora Subglacial Basin and is drained by several marine-terminating outlet



glaciers, of which the largest are Holmes (0.12 m SLE) and Frost glaciers (0.84 m SLE) (Rignot et al., 2019). Holmes Glacier
has two distinct outlets that feed the same ice shelf (Fig. 1b); the outlets are referred to as Holmes East and Holmes West
glaciers (Miles et al., 2017). In addition, a further enhanced area of flow lies between Frost and Holmes East glaciers and is
referred to as Glacier 1 in this study.

Recent studies suggest a mass imbalance from the marine-based glaciers entering Porpoise Bay, including a potential
combined sea level contribution of 0.8 mm from Holmes and Frost catchments between 1979 and 2017 (Rignot et al., 2019),
surface elevation change between $-0.07$ m yr$^{-1}$ and $-0.87$ m yr$^{-1}$ (2003−2019) (Smith et al., 2020a), ice-shelf retreat linked
to sea-ice break-out events (Miles et al., 2017), ice-shelf thinning (Smith et al., 2020a; Miles and Bingham, 2024), grounding
line retreat (Konrad et al., 2018), and ice flow speed up (Rignot et al., 2022). Despite these ice sheet-wide studies suggesting
that dynamic change may be occurring, Porpoise Bay's glaciers have not been studied in detail.

This paper aims to enhance our understanding of the recent changes in the ice dynamics of the outlet glaciers in
Porpoise Bay, East Antarctica. Following similar work undertaken in Vincennes Bay (Picton et al., 2023), also in Wilkes Land,
we assessed four key parameters of the outlet glaciers: (1) ice-shelf/glacier terminus position, delineated from satellite imagery,
(2) ice surface elevation, accessed from datasets from Schröder et al. (2019b), Smith et al. (2020b), and Nilsson et al. (2023),
(3) ice surface velocity, extracted from ITS_LIVE (Gardner et al., 2025), (4) grounding line position, assessed using differential
satellite synthetic aperture radar interferometry (DInSAR) and secondary datasets (Bindschadler and Choi, 2011; Rignot et al.,
2016; Haran et al., 2018, 2021a, 2021b). These parameters were compared to secondary data on ocean temperature from EN4
(Good et al., 2013) and MEOP casts, derived by *in situ* observations collected by marine mammals (Treasure et al., 2017), and
sea-ice concentrations from NSIDC (Fetterer et al., 2017), alongside bathymetry from Bedmap3 (Pritchard et al., 2024),
BedMachine3 (Morlighem, 2022) and ANTGG (Rignot et al., 2024) to determine the controls on dynamic changes in the outlet
glaciers draining into Porpoise Bay.



**Figure 1: (a) Ice sheet bed elevation and bathymetry of Porpoise Bay and the surrounding Wilkes land from Bedmap3 (Pritchard et al., 2024). (b) Ice velocity map of Porpoise Bay extracted from the 2021 ITS_LIVE ice velocity mosaic (Gardner et al., 2025). Central flowlines and sampling boxes are shown across the inland (IN), grounding line (GL), and floating ice tongue (FT) at each of the studied glaciers: Frost (FR), Glacier 1 (G1), Holmes East (HE), and Holmes West (HW). Black lines show the 1996 MEaSUREs grounding line (Rignot et al., 2016).**

## 2 Methodology

The recent ice dynamics of Porpoise Bay's outlet glaciers were examined by combining a range of remote sensing approaches to measure four key glacier parameters (ice-shelf position, ice surface elevation, ice surface velocity, and grounding line position) between the 1960s and the present day. This depicts the scale of change and enables assessment of the potential driving mechanisms.



## 2.1 Ice-shelf frontal position

The Google Earth Engine Digitisation Tool (GEEDiT) was used to manually map the calving front of each outlet glacier following the criteria outlined in Table 1. GEEDiT uses imagery from Landsat 4, 5, 7, 8, and 9, ASTER, and Sentinel 1 and 2, spanning from 1982 to the present day (Lea, 2018). In addition, we used co-registered and orthorectified Landsat 1 (Miles and Bingham, 2023) and declassified ARGON (Kim et al., 2007) imagery to extend our measurements back to 1963. Images were preferentially selected from austral summertime (December to February) to minimise complications with digitising sea ice. Cloud cover thresholds were not applied to avoid unnecessarily omitting images with visible ice-shelf edges. Instead, the images were manually checked to ensure the calving front was visible. The failure of the Scan Line Corrector onboard the Landsat 7 satellite resulted in striped data losses (Paul et al., 2017), but parts of the glaciers were still visible. In such images, the ice-shelf position was, where possible, digitised across gaps, using an adjacent image with a differently striped pattern of data losses to inform digitisation (Black and Joughin, 2022). The spatial resolution of imagery ranged from 140 m (Landsat 1) to 10 m (Sentinel 2).

**Table 1: Ice-shelf features and identifying criteria adapted from Holt et al. (2013) and Arthur et al. (2021). Examples of ice-shelf features from Sentinel 2 are provided.**

| Feature | Example | Structure | Identification on satellite imagery |
|---|---|---|---|
| Ice calving front (ice-free conditions) | | | Abrupt transition from ice-shelf calving margin (bright feature) to open ocean (dark feature). |
| Ice calving front (mélange presence) | | | Transition from ice shelf (smooth feature) to mélange (tightly packed calved ice blocks with other calved materials); transition often marked by shadow. |
| Ice calving front (sea ice presence) | | | Sharp transition from ice shelf ('thick' feature) to sea ice (darker than glacial ice, 'flat' feature); transition often marked by shadow. |
| Calved ice blocks/icebergs | | | Distinct blocks detached from ice-shelf edge and surrounded by mélange or open ocean |
| Mélange | | | Amalgamation of sea ice, marine ice, firn, and densely packed calved icebergs |



The outcomes of our mapping were exported to ArcGIS. The ice-shelf position change was quantified using the box method

outlined by Moon and Joughin (2008), which calculates the mean frontal change using the difference in ice-shelf geometry as measured by the area within an open-ended box across the central region of ice flow. This method accounts for asymmetric changes across glacier termini.

        The availability of cloud-free digital satellite imagery limits this method as it conceals the ice-shelf edge. When mapping the ice-shelf edge, digitisation errors were minimised by using the highest-resolution imagery available and applying

contrast stretching filters to improve the shelf-ice-mélange and ice-water boundary visibility; the resultant digitisation uncertainty was estimated to be similar to Miles et al. (2021) at 1 pixel. A particular challenge was identifying the mélange from the calving front, due to the blocky nature of the mélange. To minimise the subjectivity of our mapping this boundary, we used the feature identification criteria outlined in Table 1.

## 2.2 Ice surface elevation

Monthly surface elevation changes from 1985 to the present day were extracted from the following published ice surface elevation change datasets: Schröder et al. (2019b), Smith et al. (2020b) and Nilsson et al. (2023). The error associated with each annual velocity mosaic was provided at pixel scale, representing the standard deviation of the elevation values. Schröder et al. (2019b) calculated the monthly surface elevation changes between 1978 and 2017, relative to September 2010, at a horizontal resolution of 10 km. The associated monthly uncertainties range from ±0.3 to ±9.7 m yr$^{-1}$ (Schröder et al., 2019b).

The data were obtained from altimetry data from Seasat, Geosat, ERS-1, ERS-2, Envisat, ICESat, and CryoSat-2 satellite missions (Schröder et al., 2019b). Nilsson et al. (2023) calculated monthly surface elevation changes between 1985 and 2020, relative to December 2013, at a horizontal resolution of 1920 m. The monthly error ranges from 0.06 to 3.8 m yr$^{-1}$ (Nilsson et al., 2023). This dataset includes additional data from the ICESat-2 mission, extending the temporal coverage between 2018 and 2020.

The mean rate of elevation change was extracted from two 10 km$^2$ sampling boxes that were drawn to capture changes at key points along flow: at the grounding line (GL) and 10 km inland (IN) (Fig. 1b). Sampling from two boxes facilitates the recognition of dynamic thinning (propagation of thinning from the glacier front to the inland portion of the glacier). The measurements were calculated relative to the earliest shared time period (September 1992), to enable comparison across the datasets. We calculated the elevation anomalies relative to the 1992–2017 mean and produced a 24-month moving average for

the datasets.

        The dataset from Smith et al. (2020b) shows the rate of surface elevation change between 2003 and 2019, at a horizontal resolution of 5 km. The dataset uses altimetry data from ICESat and ICESat-2 satellite missions (Smith et al., 2020b). The dataset quantifies the root mean square error in each grid cell, which stems from instrument precision, grounding finding errors, and interpolation errors; the error from the sampling boxes ranges from 0.004 to 0.08 m. To compare the

datasets, mean rates of elevation change were calculated for Nilsson et al. (2023) between 2003 and 2019 and for Schröder et al. (2019b) between 2003 and 2017.





### 2.3 Ice surface velocities

The Making Earth Science Data Records for Use in Research Environments (MEaSUREs) ITS_LIVE product was used to extract ice surface velocity at an annual resolution between 2000 and 2022 (Gardner et al., 2025). The ITS_LIVE mosaics were derived from Landsat 4, 5, 7, and 8 imagery using autonomous Repeat Image Feature Tracking (auto-RIFT) algorithms, an open-source Python module that analyses pixel displacement between two images to calculate ice velocity (Gardner et al., 2018).

The mean annual velocities were extracted from the GL and IN sampling boxes alongside a box positioned on the floating ice tongue (FT) (Fig. 1b). Sampling from multiple boxes depicts the spatial variability in ice velocity over time to reveal any pattern or propagation of dynamic change up-ice. Ice surface velocity profiles were extracted along the central flowlines using the 2021 annual velocity mosaic for each outlet glacier and sampled at 240 m intervals to reflect the spatial resolution of the ITS_LIVE data. The 2021 mosaic was selected due to a relatively lower indicated error. The flowlines were clipped at the 2021 manually digitised ice-shelf position.

The error associated with each annual velocity mosaic was provided at pixel scale, representing the standard deviation of the difference between the image-pair component velocities and the mean annual component velocities (Gardner et al., 2018). The mean annual velocity error was extracted from each FT, GL, and IN box; the error ranged from 0.7 to 467 m yr$^{-1}$. The annual velocity data were disregarded for mean error over 50 % of the velocity magnitude, resulting in the loss of 1 % of data.

Data scarcity is a limiting factor of the early ITS_LIVE product as the auto-RIFT processing chain is limited by the availability of image pairs across a given year. Incomplete coverage was seen across Holmes East and West before the launch of Landsat 8 in 2013. The annual velocity values were disregarded if less than 25 % data coverage was observed, resulting in the removal of 8 % of values.

### 2.4 Grounding line position

Grounding line positions were derived using DInSAR interferograms from Sentinel 1 imagery collected between 2019 and 2021 (Table 2) to enable comparison with older, previously published datasets (described below). DInSAR interferograms were generated by differencing two interferograms, each produced from three or more consecutive repeat pass SAR acquisitions. Assuming constant velocity over the observed time period, the phase change from tidal flexure at the ice sheet-ice-shelf boundary can be detected from the resulting differential interferograms. These ice-shelf flexure zones are characterised by a dense fringe in the DInSAR phase.

To automatically delineate the grounding line, a Holistically-Nested Edge Detection (HED) neural network was applied to the DInSAR interferograms (Ramanath et al., 2025). The neural network was trained on the real and imaginary interferometric features, achieving a median and mean offset of 265 m and 421 m, respectively, from manual grounding line delineations, and a predictive uncertainty of 401 m (Ramanath et al., 2025). The uncertainty was derived by calculating the



pixel-wise standard deviations of each DInSAR interferogram of an ensemble of five HED neural networks, configured with
the same hyperparameters, but differing in the initialization and trained with randomly shuffled training samples (Ramanath
et al., 2025). We did not extract any coherent interferograms over Glacier 1 or Frost glaciers because the ice flow was too rapid
to get coherence within the 6-day repeat-pass interval of the Sentinel 1 constellation. At Holmes West Glacier, we identified a
2020 grounding line, but its implausibility relative to secondary datasets and very wide uncertainty buffer from a highly de-
correlated interferogram led us to disregard the line.

The grounding line positions were also extracted from several secondary datasets in the published literature. The
MEaSUREs grounding line product was generated using similar DInSAR techniques applied to ERS-1 and ERS-2 imagery
collected in 1996 (Rignot et al., 2016). This dataset has a standard error of ±100 m (Rignot et al., 2016). The Antarctic Surface
Accumulation and Ice Discharge (ASAID) grounding line dataset was produced by manually delineating the most seaward
continuous break in slope using Landsat 7 images between 1999 and 2003 and surface elevation data from ICESat satellite
mission (Bindschadler et al., 2011). This dataset has an estimated positional error of ±502 m for outlet glacier boundaries
(Bindschadler and Choi, 2011). The Mosaic Of Antarctica (MOA) grounding lines were also produced using manual
delineation of the most seaward break in slope, observed in 2004, 2009, and 2014 (Haran et al., 2018; 2021a; 2021b). The
MOA grounding line products have associated errors of ±250 m. The change in grounding line position was measured along
the central flowline from the most seaward MEaSUREs 1996 position.

The DInSAR results and MEaSUREs product represent the hinge line, an approximation of the grounding line position
(Rignot et al., 2016). Contrastingly, the ASAID and MOA datasets represent the break-in-slope; the change in ice surface at
the transition from grounding and floating ice (Bindschadler et al., 2011). The hinge line and break-in-slope are different
components of the grounding zone (the break-in-slope is observed seaward of the grounding line). Hence, caution must be
exercised when interpreting changes in grounding line position acquired from the different methods.

**Table 2: Imagery used in the DInSAR grounding line extraction.**

| Satellite | T1 | T2 | T3 |
| --- | --- | --- | --- |
| Sentinel 1 | 03/11/2019 | 09/11/2019 | 15/11/2019 |
| Sentinel 1 | 21/11/2020 | 27/11/2020 | 03/12/2020 |

**2.5 Bed and ice surface topography**

To interpret grounding line migration in the context of the wider subglacial topography, we extracted bed and ice surface
elevation profiles for the Porpoise Bay outlet glaciers. Bedrock elevation profiles were derived from BedMachine3
(Morlighem, 2022) and Bedmap3 (Pritchard et al., 2024) along the digitised flowlines. Bed elevation and the associated error
were sampled at 500 m intervals (the horizontal resolution of the dataset). Although there are 47,492 line-kilometres of airborne




radar profiles over the Aurora Subglacial Basin (Young et al., 2011), including over Porpoise Bay, the roughness and shape of the bed are not well characterised, and the presence of inland pinning points that may restrict future grounding retreat are unknown. The bed elevation error associated with Bedmap3 ranges from ±7 to ±276 m for the grounded bed and ±60 m to ±306 m for the interpolated seabed (Pritchard et al., 2024). For BedMachine3, the typical uncertainties are ±36 m in well-constrained regions (driven by radar vertical resolution); uncertainties can exceed ±200 m where data are sparse due to reliance on interpolations and can exceed ±500 m beneath floating ice where sub-ice bathymetry data are lacking (Morlighem, 2022).

Bed elevation profiles along the marine area in front of the glaciers were digitised from the AntGG2021 dataset which is based on a 3D inversion of a circumpolar compilation of gravity anomalies constrained by measurement from BedMachine3, the International Bathymetric Chart of the Southern Ocean, and discrete seafloor measurements from seismic and ocean robotic probes (Rignot et al., 2024; Charrassin et al., 2025). This dataset provides insight into the seafloor elevation of ice-shelf cavities and continental shelf. The potential error of the estimated bathymetry for the Sabrina region, where Porpoise Bay is situated, is 185 m (Charrassin et al., 2025).

Surface topography profiles were extracted along each central flowline from the Reference Elevation Model of Antarctica (REMA) mosaic (Howat et al., 2022). The surface elevation product from REMA is provided at 100 m resolution with errors of less than 1 m (Howat et al., 2019). These were sampled at 500 m intervals to match the bedrock elevation. The ice-shelf base elevation was calculated by subtracting the Bedmap3 ice thickness from the Bedmap3 ice surface elevation (Pritchard et al., 2024). The uncertainty of the Bedmap3 ice thickness dataset ranges between ±0 m for rock cells, ±10 m for ice shelves, ±10–20 m in surveyed cells, ±73–272 m in interpolated or poorly constrained cells (Prichard et al., 2025). The Bedmap3 ice surface elevation error is, on average, ±7 m (Pritchard et al., 2025).

## 2.6 Ocean and sea-ice conditions

We analyse ocean temperature and sea-ice conditions to understand their potential influence on any observed changes in ice dynamics. Due to an absence of direct, observational ocean temperature data across the continental shelf in the Wilkes Land, we used EN4.2.2 subsurface ocean temperature objective analysis data (Good et al., 2013), accessed from the UK Met Office. The data are available monthly at 1º × 1º spatial resolution. We extracted the mean monthly temperature from four cells between 65º and 66ºS, 127º and 128ºE, located on the continental shelf proximal to Porpoise Bay (Fig. 1a), for each month between January 1990 and March 2025 at 24 depths. We acknowledge that the nature of EN4 analysis data creates very high uncertainty estimates, but these data provide the only consistently derived indication of ocean temperature through time.

We supplement the EN4 data with direct measurements of ocean temperature collected by instrumental marine mammals between 2004 and 2021, available from the Marine Mammal Exploring the Oceans Pole to Pole (MEOP) Consortium (Treasure et al., 2017). Each mammal is equipped with a Conductivity Temperature Density Satellite Relay Data Logger (CTD-SRDL) and deployed for 72 to 136 days (Roquet et al., 2014; Treasure et al., 2017). Data are collected throughout a dive and once the mammal surfaces the data are processed by the CTD-SRDL sensors and telemetered through the Argo satellite system. The geolocation of each dive is extracted by the Argo satellite triangulation, and collected with an accuracy of 4 km. The data



are not spatially or temporally continuous: there are numerous casts near the edge of the continental shelf but none within 130 km of the Porpoise Bay ice margin.

Sea-ice conditions in Porpoise Bay were analysed using the NSIDC Sea-ice Index, Version 3 (Fetterer et al., 2017). The data are available at 25 km spatial resolution. We extracted the monthly sea-ice concentrations from a 12,500 km$^2$ polygon

between November 1979 and April 2025 (Fig. 1a). We recognise this region may include ice mélange and shelf ice, so is likely to capture mélange and ice-shelf breakup as well as sea-ice concentration changes. The data are obtained from satellite passive micro-wave derived datasets (the Near-Real-Time DMSP SSMIS Daily Polar Gridded Sea-ice Concentrations and the Sea-ice Concentrations from Nimbus-7 SMMR and DMSP SMM/I-SSMIS Passive Microwave Data). The accuracy of the sea-ice concentration is usually cited within ±5 % in winter and ±15 % in summer (Fetterer et al., 2017). The error is reduced for

thicker and higher sea-ice concentrations due to the reduced influence of open water or high cloud liquid on the measured brightness temperature (Cavalieri, 1995; Spreen et al., 2008; Fetterer et al., 2017).

## 3 Results

### 3.1 Ice-shelf position

The ice-shelf position of Frost, Glacier 1, Holmes East, and Holmes West glaciers fluctuated between 5.7 km and 11.3 km

over the study period (1963–2025), with no clear trend of advance or retreat, but with several major calving events, followed by advance (Fig. 2 and 3). All four glaciers advanced between 1963 and 1973, at a mean rate of +281 m yr$^{-1}$, and retreated between 1973 and 1991, at a mean rate of −386 m yr$^{-1}$, with Glacier 1 and Holmes East Glacier retreating by over ~9 km. Contrastingly, 1991 to 2001 saw an advance from Frost Glacier (+177 m yr$^{-1}$), Glacier 1 (+329 m yr$^{-1}$), and Holmes East Glacier (+545 m yr$^{-1}$) and a retreat from and Holmes West Glacier (−52 m yr$^{-1}$). However, data scarcity before 2000 is likely

to mask smaller-scale periods of advance and retreat, so these trends are evaluated with caution. We observed the ice shelves advancing to a similar maximum position within 2 km of the previous advance limit (Fig. 2), before rapidly calving and beginning to advance once more.

The ice shelves in Porpoise Bay underwent almost synchronous cycles of advance and retreat between 2006 and 2025 (Fig. 3). Initially, the glaciers advanced from a post-calving position: +5.4 km at Frost Glacier (2008–2016), +6.2 km at Glacier

1 (2007–2015), +2.7 km at Holmes East Glacier (2006–2016) (underwent another calving event between 2010 and 2011), and +6.8 km at Holmes West Glacier (2006–2016). Between 2016 and 2017, the outlet glaciers underwent a calving event, retreating −2.8 km at Frost Glacier, −8 km at Glacier 1, −0.9 km at Holmes East Glacier, and −4.5 km at Holmes West Glacier. Subsequently, Frost, Glacier 1, Holmes East and West glacier gradually advanced +0.6 km, +3.7 km, +3.1 km, and +4.2 km, respectively, between 2017 and 2021. Glacier 1, Holmes East, and Holmes West glaciers calved between 2021 and 2022 by

−2.7 km, −7 km, and −5.4 km, respectively, whilst Frost Glacier retreated −3.6 km between 2022 and 2023. Finally, the glaciers advanced +2.4 km (Frost), +1.2 km (Glacier 1), +3.7 km (Holmes East), and +0.6 km (Holmes West) by 2025.





**Figure 2: (a) Minimum and maximum ice shelf positions manually digitised across Porpoise Bay: (b) Holmes East Glacier, (c) Frost Glacier, (d) Glacier 1, and (e) Holmes West Glacier. The background satellite imagery is a Landsat image from 2024 and was downloaded from USGS EarthExplorer.**






**Figure 3: Width-averaged ice-shelf calving position changes relative to the first measurement in 1963 at (a) Frost Glacier, (b) Glacier 1, (c) Holmes East Glacier, and (d) Holmes West Glacier. Terminus positions before 2000 are joined by dashed lines due to large gaps in satellite imagery availability. Vertical dashed lines display the 2007 and 2016 sea-ice break-out event from Miles et al. (2017). Grey lines show the timing of major calving events.**






### 3.2 Ice surface velocity

### 3.2.1 Temporal variability

Temporal variations in ice surface velocity were observed between 2000 and 2022 (Fig. 4). All three glaciers saw an increase
in ice flow velocity over the study period, across parts of the glacier flowlines. Observations showed the greatest speed up at
Holmes East Glacier (Fig. 4c), with ice surface velocity at the IN box increasing +32 % overall from 382.3 ±37 in 2013 to
504 ±121 m yr$^{-1}$ in 2022. However, the absolute value of velocity increase (122 m yr$^{-1}$) is similar to the associated error. We
find significant ice surface velocity variations in the GL and IN boxes at Holmes East and West glaciers, with a period of
slowdown measured between 2017 and 2020, followed by a subsequent speed-up. Interestingly, this variation in ice surface
velocity was not translated to the floating tongue for either glacier. Glacier 1 increased to peak velocity in 2010 before slowing
down to a lower constant velocity (Fig. 4b). The ice surface velocity at Frost Glacier consistently increased +7 % to +10 %
over the study period (Fig. 4a).

### 3.2.1 Velocity and ice-shelf position

Some of the outlet glaciers in Porpoise Bay displayed coincidental timing between changes in ice-shelf position and velocity
(Fig. 4, grey). Frost Ice Shelf underwent a calving event between 2007 and 2008, after which the velocity increased +2 % to
+6 % over the following two years across the glacier. Subsequently, the ice flow velocity increased following the small retreat
between 2013 and 2014 and the ice-shelf calving between 2016 and 2019, increasing +3 % (IN), +10 % (GL), and +9 % (FT)
over 8 years (Fig. 4a). The velocity across Glacier 1 increased +18 % (IN), +13 % (GL), and +9 % (FT) in the two years after
the 2007 calving event, during which the ice-shelf area remained low, followed by a velocity slow down −24 % (IN), −7 %
(GL), and −10 % (FT) over the three years after 2010, as the ice shelf regrew (Fig. 4b). However, we do not record a velocity
response to the latter two major calving events from Glacier 1. Following the 2016 calving event, Holmes East Glacier
displayed an increase in velocity of +13 % (IN), +10 % (GL), and +5 % (FT) between 2016 and 2017 (Fig. 4c). The glacier
showed limited velocity response to the 2021 calving event; the velocity continued its upward trend. Holmes West Ice Shelf
underwent a calving event between 2007 and 2008, during which a +6 % increase in velocity was recorded across the grounding
line (Fig. 4d). The subsequent 2016–2017 calving event is coincidental with a +3 % increase in velocity across the floating
tongue. Conversely, the subsequent 2021 calving event is followed by a −5 % decline in ice velocity across the floating tongue
and further inland. Overall, the outlet glaciers display some changes in velocity that correspond to changes at the ice-shelf
edge, but we do not record a consistent response to every calving event.




**Figure 4: Mean annual velocity extracted from the inland (IN), grounding line (GL), and floating ice tongue (FT) boxes across (a) Frost Glacier, (b) Glacier 1, (c) Holmes East Glacier, and (d) Holmes West Glacier. Velocity data extracted from the ITS_LIVE velocity mosaic between 2000 and 2022 (Gardner et al., 2025). Dashed lines shows the linear trend. Grey lines show the timing of major calving events at the glacier ice shelf; grey dashes lines depict a partial/minor calving event. The y-axis scales are different across Figure 4.**



### 3.3 Ice surface elevation change

There was general agreement between the two ice surface elevation datasets provided by Nilsson et al. (2023) and Schröder et al. (2019b) across the GL and IN boxes, especially after *ca.* 2003, when the uncertainties also decrease. Indeed, the datasets are associated with high levels of uncertainty in the 1990s and early 2000s. Across the IN boxes, a clear and consistent trend of thinning was observed across all four glaciers, with Holmes East and West thinning more rapidly than Frost and Glacier 1 (Fig. 5). Between 2003 and 2017, Schröder et al. (2019b) and Nilsson et al. (2023) observed thinning at an average rate of $-0.41$ and $-0.35$ m yr$^{-1}$ for Holmes East Glacier and $-0.04$ and $-0.34$ m yr$^{-1}$ for Holmes West Glacier (values relative to 09/1992). Contrastingly, Frost Glacier and Glacier 1 displayed very little change in ice surface elevation between 2003 and 2017, with thinning rates calculated at an average of $-0.01$ and $0.003$ m yr$^{-1}$ for Frost and $-0.16$ and $-0.01$ m yr$^{-1}$ for Glacier 1 from Schröder et al. (2019b) and Nilsson et al. (2023).





Figure 5: **Monthly surface elevation anomalies observed in each IN box at (a) Frost Glacier, (b) Glacier 1, (c) Holmes East Glacier, and (d) Holmes West Glacier between 1992 and 2020. Elevation anomalies are calculated relative to the 1992–2017 mean. Bold lines display the 24-month rolling mean.**




Mirroring the elevation patterns inland, observations of surface elevation across the GL boxes show rapid thinning at Holmes East and West Glaciers (Fig. 6). Schröder et al. (2019b) and Nilsson et al. (2023) observed thinning at an average rate of −0.41 and −0.40 m yr$^{-1}$ for Holmes East Glacier and Nilsson et al. (2023) observed average thinning at a rate of −1.22 m yr$^{-1}$ for Holmes West Glacier between 2003 and 2017. We observed lower rates of surface elevation loss at Frost Glacier and Glacier

1. Schröder et al. (2019b) and Nilsson et al. (2023) observed thinning at an average rate of −0.08 and −0.13 m yr$^{-1}$ for Frost Glacier and −0.17 and 0.001 m yr$^{-1}$ for Glacier 1.





Figure 6: Monthly surface elevation anomalies observed in each GL box at (a) Frost Glacier, (b) Glacier 1, (c) Holmes East Glacier, and (d) Holmes West Glacier between 1992 and 2020. Elevation anomalies are calculated relative to the 1992-2017 mean. Bold lines display the 24-month rolling mean.






Similar trends are observed by Smith et al. (2020b) using ICESat- and ICESat-2-derived data (Fig. 7). Porpoise Bay showed an overall decrease in ice surface elevation between 2003 and 2019, with a hotspot of thinning extending across the ice shelf and grounded ice at Holmes East and West glaciers. We observed a maximum surface elevation loss of −1.11 m yr$^{-1}$ on the grounded ice and −3.07 m yr$^{-1}$ on the floating ice shelf.


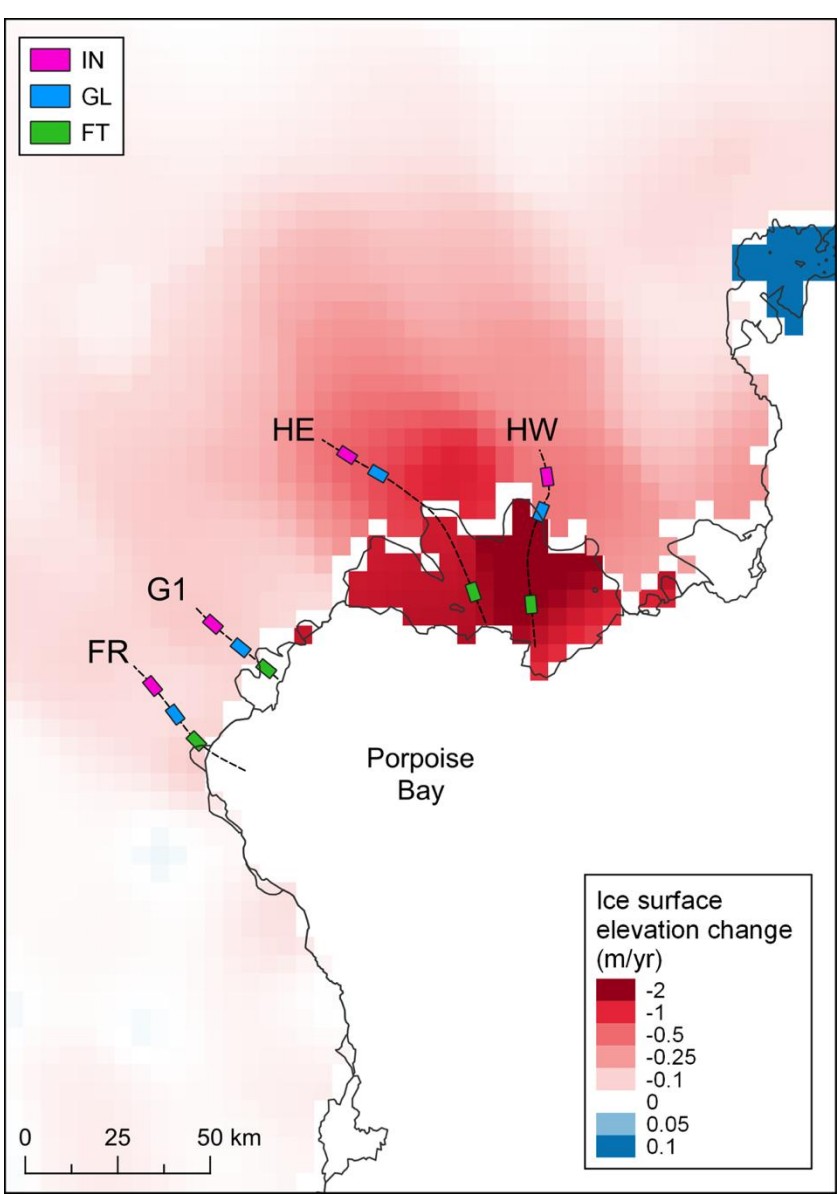

**Figure 7: Rate of ice-surface elevation change observed inland of Porpoise Bay between 2003 and 2019, calculated by Smith et al. (2020b).**



**3.4 Grounding line position and bathymetry**

Notwithstanding some very large uncertainties (see Section 4.1), the highest grounding line retreat was observed at Frost and Holmes East glaciers (Fig. 8 and 9). If the MEaSUREs data from 1996 are taken at face value, Frost Glacier retreated −17.17 km between 1996 and 2014, at an average rate of −0.95 km yr$^{-1}$ (Fig. 8a). This grounding line retreat occurred largely between 1996 and 2004, at an average rate of −2.14 km yr$^{-1}$, after which the grounding line was observed at a constant position. The

grounding line retreat occurred across a largely retrograde slope, dropping in elevation from −615 m (Bedmap3)/ −587 m (BedMachine3)/ −758 m (ANTGG2021) to −982 m (Bedmap3)/ −875 m (BedMachine3), before becoming stabilised on a bedrock pinning point (Fig. 9a).

The grounding line position at Holmes East was observed to retreat −15.24 km between 1996 and 2001, at an average pace of −3.05 km yr$^{-1}$ (Fig. 8c), where it stabilised for the following 13 years, with ASAID, and MOA datasets digitised at

nearly identical locations. However, comparing across the DInSAR-derived grounding line positions, we found −3.42 km of retreat from 1996 (MEaSUREs) to 2019, at an average rate of −0.15 km yr$^{-1}$. We digitised the 2019 position within a 10.0 km wide buffer zone, indicating retreat within the range of −9.69 to 0.34 km, relative to 1996. The retreat continues to 2020; our findings show −5.27 km of retreat from the 1996 position, calculated at −0.22 km yr$^{-1}$, with a maximum of −14.65 km and minimum of −2.75 km within the 11.9 km buffer zone, at a mean rate of −0.61 km yr$^{-1}$ to −0.11 km yr$^{-1}$. The grounding line

retreated initially down a retrograde slope before stepping up a prograde slope (Fig. 9c). Taken at face value, we observed the grounding line to retreat between 1996 and 2001, remain stable between 2001 and 2014, and then advance to 2019. The difference in grounding line position between the DInSAR and the ASAID and MOA products in Figure 8c highlights the uncertainty associated with comparing products delineating the hinge line from DInSAR and break-in-slope from manual delineation.






**Figure 8: Changes in grounding line position measured relative to the minimum 1996 position observed at (a) Frost Glacier, (b) Glacier 1, (c) Holmes East Glacier, and (d) Holmes West Glacier. The 1996 position from MEaSUREs (Rignot et al., 2016), 2001 from ASAID (Bindschadler & Choi, 2011), 2004, 2009, and 2014 from MOA (Haran et al., 2018; 2021a; 2021b), 2019 and 2020 from this paper.**




In contrast, the grounding line position of Holmes West Glacier was observed to be relatively stable between 1996 and 2014; the grounding lines were digitised at very similar positions across products (Fig. 8d). Holmes West Glacier retreated −1.28 km between 1996 and 2014, recorded at an average rate of −0.07 km yr$^{-1}$. The grounding line retreated down a very shallow slope but is positioned at the edge of a significant steepening retrograde bed that continues down a reverse slope for 20 km, with the bed lowering almost 700 m along only 2 km of grounding line retreat across the flowline (Fig. 9d).

The grounding line position at Glacier 1 showed retreat then advance (Fig. 8b). Between 1996 and 2004, the grounding line position retreated −6.01 km, calculated at an average rate of −0.75 km yr$^{-1}$. The grounding line was subsequently observed to advance +4.18 km by 2014, at an average rate of +0.42 km yr$^{-1}$. Observations show an overall retreat of −1.83 km across a largely flat bedrock elevation (Fig. 9b).





**Figure 9:** Ice surface and bed elevation profiles extracted along the central flowlines of (a) Frost Glacier, (b) Glacier 1, (c) Holmes East Glacier, and (d) Holmes West Glacier. Bed elevation: Bedmap3 (Pritchard et al., 2024), BedMachine3 (Morlighem, 2022), and ANTGG (Rignot et al., 2024), ice surface elevation: REMA (Howat et al., 2022), ice shelf base: Bedmap3 (Pritchard et al., 2024). The red lines depict the oldest and newest grounding line (dashed red in (c) shows the maximum uncertainty). Measurements are shown in the along-flow direction, in which 0 km represents the inland start point.



## 3.5 Variability in ocean and sea-ice conditions

Although there is substantial variability in EN4 ocean temperature over time (Fig. 10), these reanalysis data show warm ocean
water on the continental shelf in Porpoise Bay. At the surface (shallower than 100 m depth), we record temperatures largely
below −1°C, calculated at a mean temperature of −1.4°C between 1990 and 2025 (Fig. 11). There is annual or biennial seasonal
warming to >0°C at depths above 50 m that last for a few summer months (Fig. 10). At intermediate depths (100 to 350 m
depth), we calculate a mean ocean temperature of −0.7°C between 1990 and 2025 (Fig. 11), with seasonal cycles displaying
approximately −0.5°C to −1°C in winter and 0°C to 0.5°C in summer (Fig. 10). At depth below 450 m, the ocean is largely
warmer than 0°C with a seasonal cycle between −0.5°C (winter) and 0.5°C or warmer (summer). We calculate a mean
temperature of 0.1°C for depths below 447 m between 1990 and 2025. We record cooler periods, during which temperatures
warmer than 0°C are not present at depths greater than 500 m across 1997–1999, 2004–2005, 2007, 2011–2015, and 2019–
2021 (Fig. 10). We observe no significant trend in temperature over the 1990–2025 period (Fig. 11).

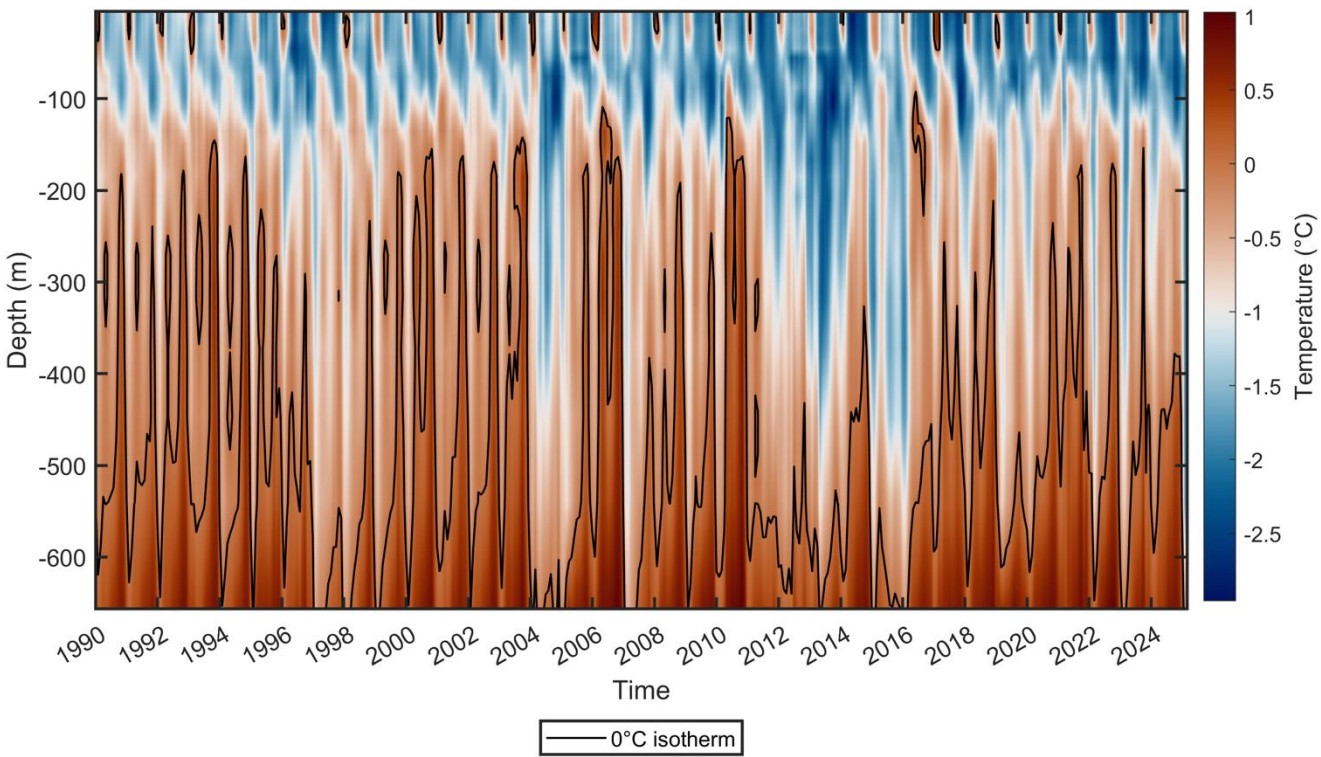

**Figure 10: EN4 subsurface ocean temperature depth profile for 1990 to 2025 from 4 grid cells between 65º and 66ºS, 127º and 128ºE
on the continental shelf, Porpoise Bay (Fetterer et al., 2017).**



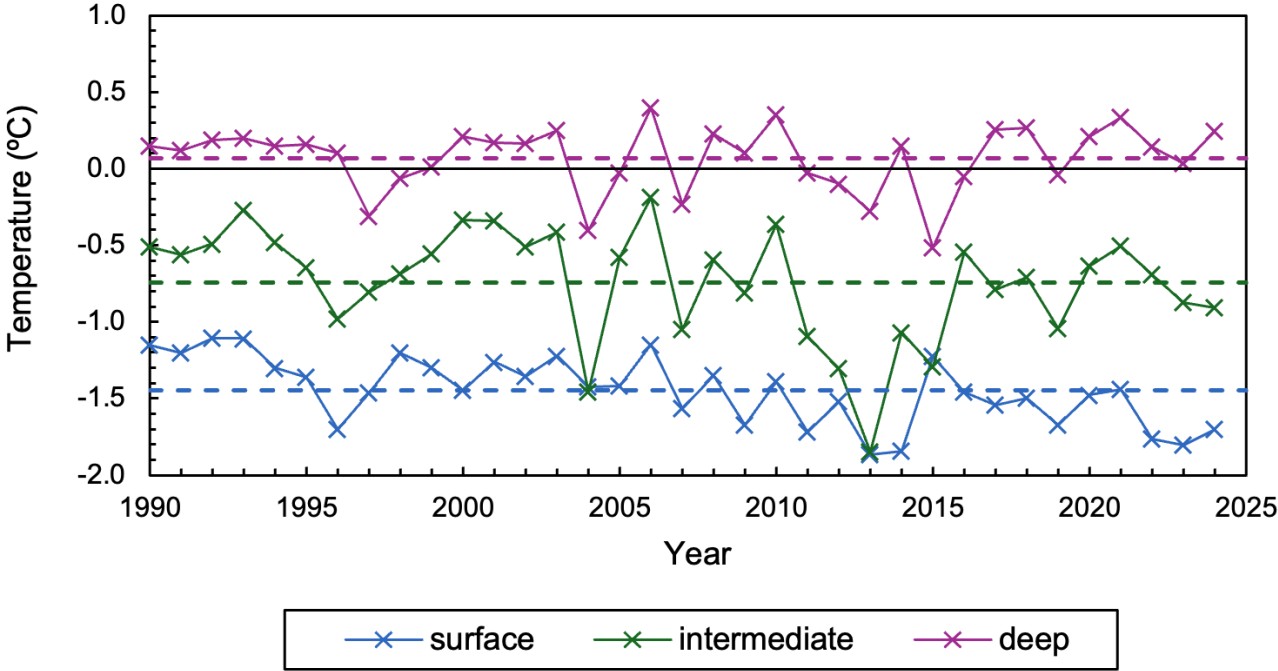

**Figure 11: Annual EN4 subsurface ocean temperature across three depth ranges (surface: above 98 m, intermediate: between 110 m and 373 m, deep: below 447 m). Dashed lines show mean temperature at each depth.**

18 MEOP casts recorded between 2004 to 2021 reveal warm, salty water between 200 m and 1000 m depth at the continental shelf break, close to Porpoise Bay (Fig. 12). Further offshore, warm, salty waters are recorded over the continental slope, ranging from 0°C to 2°C (Fig. 12a) with salinity of around 34.8 g kg$^{-1}$ (Fig. 12b) between 200 m and 1000 m depth (Fig. 12c). Over the continental shelf there are relatively few MEOP casts, with none within 130 km of the ice sheet margin, possibly due to the persistence of sea ice in the region. However, *in situ* ocean temperatures on the outer continental shelf and shelf break, north-west of Porpoise Bay, are recorded above freezing at depth, with temperatures ranging from 0.19°C (at 1000 m depth) to 0.13°C (at 918 m depth) to 0°C (at 690 m depth) (Fig. 12a and 12c). Closer to the surface, we record cooler temperatures −1.77°C (at 152 m depth) to −1.68°C (at 324 m depth). The casts record salinity up to 34.7 g kg$^{-1}$ on the continental shelf, with many of the values above 34.5 g kg$^{-1}$ (Fig. 12b). mCDW in East Antarctica is characterised by high salinity (>34.5 g kg$^{-1}$) and temperature above freezing (Schodlok et al., 2016). Our findings suggest the presence of mCDW around the continental shelf break near Porpoise Bay at depths at, or shallower than, the continental shelf edge.





**Figure 12: Oceanic properties derived from the MEOP casts between 2004 and 2021 near Porpoise Bay: (a) maximum conservative temperature (ºC) observed below 150 m depth at each seal dive, (b) absolute salinity (g kg⁻¹) at the depth of (a), (c) depth of (a), (d) bathymetry of the continental shelf from ANTGG (Rignot et al., 2024). Dashed line shows continental shelf break from Amblas (2018) at approximately 600–800 m, solid line shows ice sheet and shelf edges from MEaSUREs (Mouginot et al., 2017).**



Our analysis of monthly sea-ice concentration from 1979 to 2025 reveals lower-than-average March sea-ice concentrations in 2006, 2007, 2010, 2016, 2021, and 2022 (Fig. 13), with very low sea-ice concentrations recorded in 2007 (38 %), 2010 (50 %), and 2021 (41 %). These abnormally low March sea-ice conditions coincide with the beginning of calving events recorded at the ice shelves (Fig. 3). Furthermore, the results show lower-than-average mean sea-ice concentration in

the winter-spring (June to November) of 2006, 2015, and 2020, and the summer-autumn (Dec–May) of 2007, 2016, and 2021 (Fig. 14). These low seasonal sea-ice conditions coincide or directly precede the calving events recorded above at Frost, Glacier 1, and Holmes West ice shelves (Fig. 3).

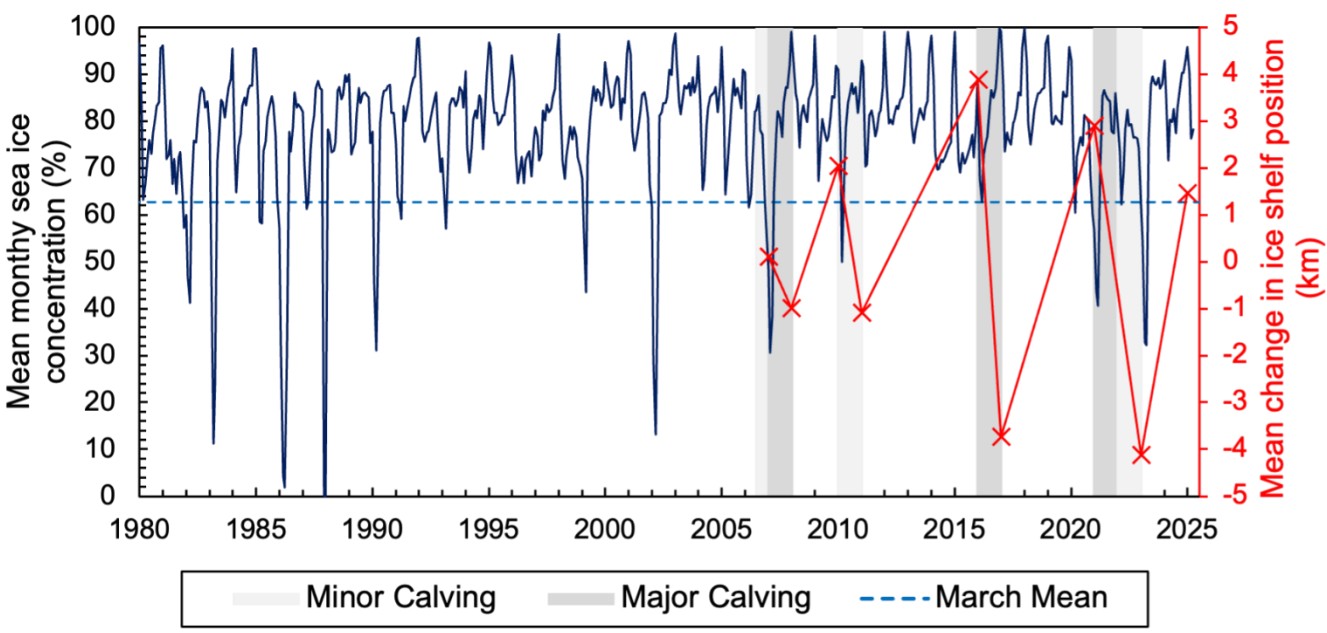

**Figure 13: Monthly sea-ice conditions extracted from a 100 km × 125 km box extending 130 km offshore from Porpoise Bay (see Fig. 1a). The March mean is shown in dashed blue. Pale grey depicts when 1 glacier underwent a calving event, dark grey denotes when 3 or more glaciers calved (see Fig. 3).**



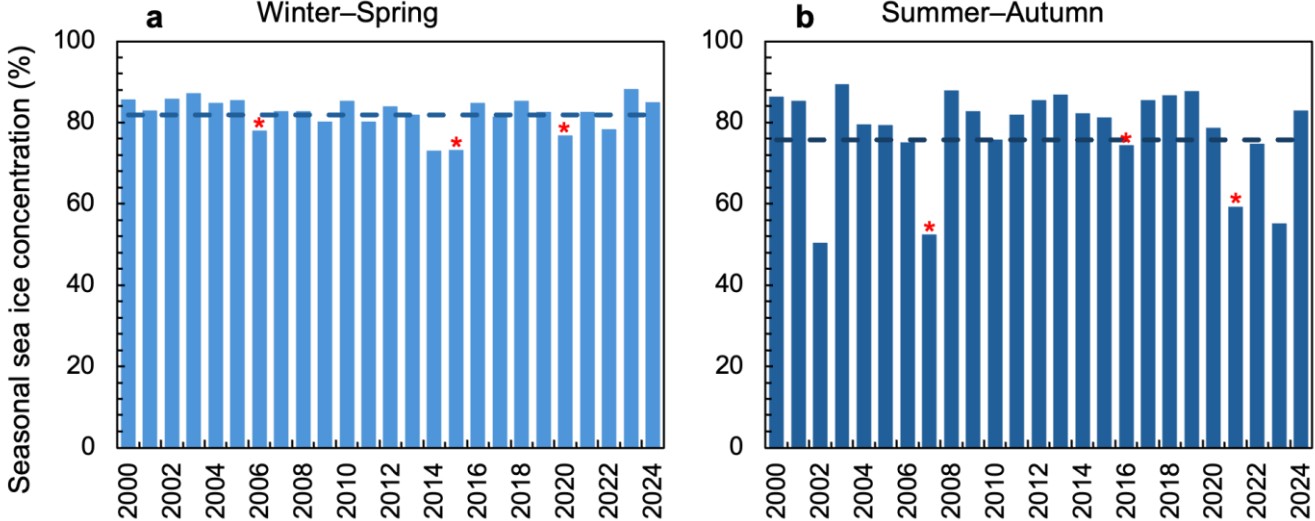

**Figure 14: Seasonal sea-ice conditions offshore of Porpoise Bay (see Fig. 1a). (a) Winter–Spring (June to November). (b) Summer–**
**Autumn (December to May). Asterisks show the key years relating to calving events at the ice-shelf fronts (Fig. 3). Dashed lines show the mean seasonal sea-ice concentration.**

## 3.6 Summary of observations

Our key observations show changes at Frost, Holmes East and Holmes West glacier that are consistent with dynamic changes
and mass loss observed elsewhere in Antarctica, particularly in West Antarctica (Table 3). Over the study period, we record grounded ice surface lowering, grounding line retreat, ice surface velocity increase, and ice-shelf retreat of varying degrees across Frost, Holmes East, and Holmes West glaciers. The highest surface elevation thinning is seen at Holmes West ($-1.22$ m yr$^{-1}$), where the grounding line retreat is modest ($-70$ m yr$^{-1}$), but where it sits on a deep bed with a retrograde bed-slope further inland. The surface elevation lowering at Frost and Holmes East is more moderate ($-0.13$ m yr$^{-1}$ and $-0.4$ m yr$^{-1}$,
respectively) but the grounding line retreat of both glaciers is >500 m yr$^{-1}$, which is high compared to most other outlet glaciers in East Antarctica (Konrad et al., 2018; Stokes et al., 2022; Picton et al., 2023). The long-term trend of ice velocity is also increasing across all three of these glaciers, albeit very slowly, with their ice front position also retreating. In contrast, the much smaller Glacier 1 shows limited evidence of dynamic change (Table 3).






**Table 3: Summary statistics of each outlet glaciers. (a) Mean annual surface elevation change across the GL between 2003 and 2017 from Nilsson et al. (2023). (b) Mean annual grounding line position change between 1996 and 2014 or 2020 (maximum position within the buffer). (c) Mean annual percentage change in ice surface velocity between 2000 and 2022 (2014–2022 for HE and HW). (d) Mean annual ice-shelf calving front change between 1963 and 2025. (e) Change in bed elevation between the 2014/2020 grounding line position and 10 km inland along the flowline from Bedmap3. Formatting indicates the degree to which each statistic reflects dynamic change: values in *bold italics* show the strongest evidence of dynamic change, values in *italics* show moderate evidence, and values in** plain test **show minimal or inconsistent evidence.**

|  |  | Frost Glacier | Glacier 1 | Holmes East Glacier | Holmes West Glacier |
|---|---|---|---|---|---|
| a. | surface elevation | *− 0.13 m yr⁻¹* | 0 m yr⁻¹ | *− 0.4 m yr⁻¹* | ***− 1.22 m yr⁻¹*** |
| b. | grounding line position | ***− 950 m yr⁻¹*** | *− 100 m yr⁻¹* | ***− 610 m yr⁻¹*** | *− 70 m yr⁻¹* |
| c. | ice surface velocity | + 0.3 % | − 0.5 % | *+ 1.7 %* | + 0.1 % |
| d. | ice-shelf position | *− 110 m yr⁻¹* | *− 17 m yr⁻¹* | − 3 m yr⁻¹ | *− 29 m yr⁻¹* |
| e. | bed elevation | − 4 m | + 18 m | + 460 m | ***− 783 m*** |

## 4 Discussion

### 4.1 Grounding line retreat and ice surface thinning

Porpoise Bay displays a pattern of localised grounding line retreat. Taken at face value, our results would make Frost one of the fastest retreating glaciers in East Antarctica on decadal timescales at an average rate of −0.95 km yr⁻¹ between 1996 and 2014 (Fig. 8a), down a retrograde slope (Fig. 9a), with a more rapid rate of −2.15 km yr⁻¹ between 1996 and 2001 (Fig. 8a). Neighbouring Holmes East Glacier underwent grounding line retreat at a maximum rate of −0.61 km yr⁻¹ between 1996 and 2020 (Fig. 8c), observed at the maximum width of the grounding line uncertainty buffer zone. For context, only Vanderford Glacier in East Antarctica sustained similarly high rates of grounding line retreat at −0.78 km yr⁻¹ (1996–2020) (Picton et al., 2023). In the WAIS, Thwaites and Pine Island glaciers in the Amundsen Sea Embayment experienced similarly rapid rates of −0.8 km yr⁻¹ and −0.95 km yr⁻¹ between 1992 and 2011 (Park et al., 2013; Milillo et al., 2019).

We acknowledge that our measurements of grounding line retreat at Frost Glacier are highly uncertain due to a comparison across products that not only delineate different parts of the grounding zone (i.e., hinge line vs break-in-slope), but also use different methods (i.e., DInSAR vs manual delineation). Furthermore, the difference between the DInSAR and the ASAID and MOA positions at Holmes East Glacier (Fig. 8c) highlight the uncertainty between the ASAID and MOA grounding line position results at Frost Glacier (Fig. 8a), suggesting retreat could have been less rapid than we record. However, even if the rates of retreat are not as high as we report in Figure 8a (due to uncertainties), our finding are broadly consistent with the grounding line retreat observed by Konrad et al. (2018), measured at −0.2 km yr⁻¹ between 2010 and 2016 using



satellite altimetry and ice geometry, suggesting that grounding line retreat is almost certainly occurring. Further identification
of more recent and accurate grounding line positions at Frost Glacier are therefore a priority.

Substantial grounded ice surface elevation lowering is observed at Holmes East and West glaciers, propagating inland
across datasets (Schröder et al., 2019b; Smith et al., 2020b; Nilsson et al., 2023), indicating a potential signal of dynamic
thinning. Holmes West Glacier displayed thinning up to −1.22 m yr⁻¹ at the grounding line (Fig. 6d), propagating 10 km inland
at a rate of −0.34 m yr⁻¹ (Fig. 5d). This rapid rate places Holmes West Glacier as one of the most rapidly thinning glaciers in
East Antarctica over decadal timescales (Stokes et al., 2022); only at Totten Glacier are there observations of such extreme
thinning, measured up to −1.9 ± 0.07 m yr⁻¹ (2003–2007) (Pritchard et al., 2009), −1.22 ± 0.1 m yr⁻¹ (2003–2019) (Smith et
al., 2020b) and −0.72 ± 0.02 m yr⁻¹ (2010–2019) (Li et al., 2023). Our results also suggest that Frost, Holmes East, and West
glaciers thinned whilst ice surface velocity increased modestly (Fig. 4), suggesting a connection between flow regime and ice
thickness that is consistent with dynamic thinning (Pritchard et al., 2009).

Further analysis suggests a relationship between grounding line retreat and ice surface thinning rates in Porpoise Bay
across our study period. At Holmes East and West glaciers, the most rapid grounding line retreat took place between 1996 and
2001 (Fig. 8: −15.24 m and −1.81 m), during which we recorded enhanced thinning rates (Fig. 6: −0.63 m yr⁻¹ and −2.29 m
yr⁻¹). This correlates with a pan-ice sheet study by Konrad et al. (2018) that found an approximately proportional relationship
between ice thickness change and grounding line migration within sections of ice flowing faster than 800 m yr⁻¹. Overall, the
extreme localised grounding line retreat and the dynamic thinning, alongside ice surface velocity speed-up, are consistent with
a sustained period of change in Porpoise Bay that is indicative of a dynamic response to external forcing.

**4.2 Role of ocean forcing in Porpoise Bay glacier dynamics**

High rates of grounding line retreat and ice surface lowering inland have been widely attributed to the intrusion of warm
mCDW across the continental shelf towards Antarctic sub-ice cavities (Thoma et al., 2008; Paolo et al., 2015; Scambos et al.,
2017; Rignot et al., 2019). The rates of grounding line retreat observed at Frost Glacier and Holmes East Glacier are consistent
with warm mCDW incursion beneath Frost Ice Shelf and Holmes East Ice Shelf. Although the EN4 data do not depict a long-
term warming trend (Fig. 11), recent observations of mid-depth CDW along the continental slope off East Antarctica show
warming since the 1990s, which is attributed to the southwards shift in the Antarctica Circumpolar Current (ACC), potentially
driven by a poleward shift in the westerlies over the Southern Ocean that is linked to a positive trend in the Southern Annular
Mode (Yamazaki et al., 2021; Herraiz-Borreguero and Garabato, 2022).

We recorded 18 *in situ* temperature and salinity measurements from MEOP casts that indicate mCDW presence on
the outer continental shelf and shelf break, and CDW offshore of the continental shelf between 2004 and 2021 (Fig. 12). In the
absence of ocean temperature observations within 130 km of the ice-shelf margin, we use EN4 subsurface ocean temperatures
to gain potential insight into the existence of mCDW across the continental shelf between 1990 and 2025 (Fig. 10). Though
we acknowledge that the nature of these data creates high uncertainty estimates, EN4 data suggest the presence of mCDW at
depth across the continental shelf, with an average ocean temperature of 0.7°C below 447 m, and a seasonal cycle of 0–0.5°C



at intermediate depth (100–350 m) between 1990 and 2025. The intrusion of mCDW has the potential to cause ice-shelf melting and thinning, delivering freshwater that could inhibit the formation of dense shelf water in Porpoise Bay, strengthening water column stratification and enabling the enhanced intrusion of warm mCDW at depth (Ribeiro et al., 2021).

At Holmes Ice Shelf (East and West), high rates of ice-shelf basal melt have been estimated, which is consistent with intrusions of mCDW. Using satellite radar altimetry with satellite-derived ice velocities and a model of firn-layer evolution, Adusumilli et al. (2020) calculated an average basal melt rate of $13.3 \pm 2.9$ m yr$^{-1}$ beneath the Holmes Ice Shelf between 1994 and 2018, the highest average basal melt rate of any ice shelf in East Antarctica, and fourth in Antarctica behind the ice shelves of Thwaites ($26.7 \pm 2.4$ m yr$^{-1}$), Land ($20.4 \pm 2.7$ m yr$^{-1}$), and Pine Island ($14.0 \pm 1.6$ m yr$^{-1}$) glaciers. Adusumilli et al. (2020)

reported that Holmes Ice Shelf had a higher average basal melt rate than Totten Ice Shelf ($11.5 \pm 2.0$ m yr$^{-1}$), where hydrographic observations show warm mCDW at the ice front (Rintoul et al., 2016).

         Studies also show that Holmes Ice Shelf has been thinning considerably. Using ICESat- and ICESat-2-derived data, Smith et al. (2020b) recorded a maximum ice surface lowering rate of $-3.07$ m yr$^{-1}$ (2003–2019) on the floating ice shelf. A recent study extended this thinning record using the surface expression of pinning points as a proxy for ice-shelf thickness;

Miles and Bingham (2024) found that Holmes Ice Shelf began to thin at least 50 years ago, alongside Moscow University and Totten, evinced by a reduction in pinning across three study epochs: 1973–1989, 1989–2000, and 2000–2022. Miles and Bingham (2024) concluded by noting that Holmes Ice Shelf was showing signs of acceleration and pinning-point loss and the continuation of pinning-point loss would likely reduce buttressing and cause acceleration of ice discharge and mass loss.

         Porpoise Bay's pattern of localised rapid grounding line retreat and thinning is consistent with a dynamic response of

glaciers flowing into thinning ice shelves due to warm mCDW intrusion. To enable mCDW to access the grounding line, sufficiently deep cross-shelf troughs must exist in front of Porpoise Bay to transfer warm water from the continental shelf break to the grounding line. Although bathymetric data are limited across Porpoise Bay, due to the persistence of sea ice in the bay, a recent study using seal data found several dives at the edge of the continental shelf, offshore from Porpoise Bay, that were up to 500 m deeper than the reported bathymetry (McMahon et al., 2023). Indeed, a trough may exist that connects the

open ocean and grounding line via a deeper corridor of topography across the continental shelf that could transport mCDW (Fig. 12d). In addition, the ice-shelf cavity must be sufficiently deep for mCDW to access the grounding line. Using 3D inversion, seismic, and MBES/SBES data, Rignot et al. (2024) found the bed at the ice margin of all four outlet glaciers was deeper than reported by BedMachine3 or Bedmap3, by over 200 m for several tens of kilometres along the flowlines (Fig. 9 and 13d). Visual inspection of the dataset (Rignot et al., 2024) reveals deep sub-ice-shelf cavities below Frost, Holmes East,

and Holmes West glaciers, between 650 m to 750 m depth, that deepen inland (Fig. 9). These cavities are sufficiently large to plausibly provide a pathway for mCDW, which the EN4 and MEOP datasets suggest may exists below approximately 450 m depth (Fig. 10), to circulate under the ice shelves and melt the base of the ice shelves, consistent with satellite-derived high basal melt rates (Adusumilli et al., 2020) and ice-shelf thinning (Smith et al., 2020b; Miles and Bingham, 2024) reported elsewhere and likely causing the dynamic changes observed in Porpoise Bay.



The bed topography beneath the outlet glaciers may differentially enhance the impact of ocean forcing in Porpoise Bay. We record sufficiently low bed elevation across the observed grounding line positions at Frost (Fig. 9a: below 600 m in 1996 and 900 m in 2014) and Holmes East (Fig. 9c: below 800 m in 1996 and 900 m in 2020) for mCDW to plausibly access the grounding line, driving grounding line retreat. Another key detail revealed by the ANTGG2021 dataset is that Holmes West Glacier has been grounded on a bedrock high between 1996 and 2014 (Rignot et al., 2024). The bed seaward of the

grounding zone ranges from 600 m to 740 m depth but rises to 500 m to 485 m depth at the grounding zone (Fig. 9d). This bathymetry may limit warm mCDW from access the grounding line, which may explain the lower rate of grounding line retreat at Holmes West Glacier, compared to Frost and Holmes East glaciers (Fig. 8), whilst also explaining the rapid surface lowering observed (Fig. 6d), since Holmes West Glacier may flow into a rapidly thinning ice shelf, potentially caused by basal melting. Furthermore, the bed at Glacier 1 is entirely above 500 m depth, which may limit potential exposure to mCDW at the grounding

line, explaining the lack of observed dynamic change at Glacier 1 (Table 3).

## 4.3 Role of sea ice in driving ice shelf and glacier flow changes

Variations in sea-ice conditions likely influenced the behaviour of calving events, ice-shelf area and ice flow in Porpoise Bay. The ice shelves underwent near-simultaneous calving events over the study period; observations show calving events in 2006–2008, 2016–2017, and 2021–2022. Recent studies around Antarctica demonstrate that the presence of sea ice and mélange

may exert a resistive backstress on the ice-shelf calving front, and calving events have been linked to the removal break-up of sea ice and the removal of mélange from the ice front (Amundson et al., 2010; Miles et al., 2017; Arthur et al., 2021; Baumhoer et al., 2021; Gomez-Fell et al., 2022; Kondo and Sugiyama, 2023; Parsons et al., 2024). Resultant ice-shelf calving can reduce their buttressing potential (Dupont and Alley, 2005; Fürst et al., 2016), which can trigger dynamic thinning and ice flow acceleration across the glacier further upstream (Reese et al., 2018; Gudmundsson et al., 2019).

Our findings show that all three major calving events appear to coincide with lower-than-average winter-spring sea-ice concentrations the preceding year and lower-than-average summer-autumn sea-ice conditions at the beginning of each major calving event (2006–2008, 2016–2017, and 2021–2022) (Fig. 14). Furthermore, the monthly sea-ice concentration was anomalously low in March 2007 and March 2021 (Fig. 13). This is supported by our observations from Landsat 7 and Landsat 8 satellite imagery that shows the break-up and removal of the mélange and sea ice adjacent to Holmes Ice Shelf prior in 2007,

2017 and 2021. Therefore, it is likely that the break-up of sea ice and the removal of iceberg mélange reduced the stabilising backstress on the ice-shelf fronts in Porpoise Bay, culminating in major, near-synchronous calving events. This is consistent with Miles et al. (2017), who attributed the 2007 and 2016 calving events to the break-up of multi-year sea ice and suggested that atmospheric circulation anomalies caused the sea-ice break-up. We also observed abnormally low sea-ice conditions in 2002 (Fig. 13), when we also recorded a calving event from Holmes East Ice Shelf (there are no positional record for the other

ice shelves due to a lack of cloud-free imagery).

Porpoise Bay's ice shelves advanced and retreated cyclically, each time advancing to a similar (±1 km) maximum length prior to calving (Fig. 2). In a previous study of the ice shelves in Porpoise Bay, Miles et al. (2017) observed Holmes



West Ice Shelf calving at the same position in each cycle and found that this took place as the glacier advanced and pushed the sea ice (attached to the ice-shelf calving front) out of the bay towards the open ocean, causing it to disintegrate. It is likely that this mechanism of ice shelves bulldozing the sea ice operated at all four glaciers and was in part facilitated by the winter-spring sea-ice conditions from the preceding years.

Our findings suggest that ice-shelf calving in Porpoise Bay, alongside the intermittent loss of stabilising sea ice and mélange, had some corresponding impact on ice flow velocity for the outlet glaciers, suggesting that some of the ice shelves buttressed ice flow. This is consistent with Fürst et al. (2016), who used an ice flow model to calculate that only 18.3 % of the ice-shelf area in Porpoise Bay was passive (i.e., has no dynamical influence on ice discharge), meaning that once calving exceeds this 18.3 % of ice-shelf area, further ice shelf calving will have a dynamic impact inland. The most consistent velocity response followed the 2007 calving event (Fig. 3), after which Frost, Glacier 1, and Holmes West glaciers displayed an increased velocity response over the following few years (Fig. 4). Given that the calving event coincided with low sea-ice conditions (Fig. 13 and 14), these findings suggest a sea-ice-driven reduction in buttressing force, indicating that the ice shelves calved beyond the passive-ice area (Fürst et al., 2016). The debuttressing velocity response to the 2016 calving event was more varied, with an increase in velocity observed between 2016 and 2017 at Holmes East and West glaciers (Fig. 4c and 4d). Given the ice-shelf retreat was relatively small at Holmes East Glacier, this debuttressing is likely caused by the sea-ice break-up and low sea-ice concentration leading up to the calving, rather than the loss of buttressing ice-shelf area. For Frost Glacier, the onset of flow speed-up coincided with a small calving event in 2013, after which the ice shelf broadly retreated, suggesting that the loss of back-stress exerted by the ice shelf is a potential driver of glacier speed-up. During high sea-ice concentrations between 2017 and 2020 (Fig. 14b: summer-autumn mean calculated 10 % above average) and the readvance of the ice shelves (Fig. 3), we record a decrease in ice flow velocity, with Holmes East and West slowing −32 % and −5 % across the grounding line and −20 % and −28 % across the inland ice (Fig. 4). This suggests the glacier flow speeds were reduced by the accumulation of sea ice and ice mélange near the glacier front. Subsequently, between 2020 and 2021, we record autumn–winter and winter–spring sea-ice concentration below average, ice-shelf retreat, and ice flow velocity increase across all sampling boxes in Holmes East and West glaciers, suggesting that the low sea-ice conditions impacted glacier dynamics before the 2021/22 ice-shelf calving event. It is difficult to assess the impact of the 2021/22 calving event on glacier flow dynamics and the impact of sea-ice-driven debuttressing that occurred due to the lack of ice flow velocity data beyond 2022.

In the future, we may see an increase in low sea-ice states that could impact the future ice-shelf stability in Porpoise Bay, as low sea-ice conditions continue to destabilise ice shelves, and drive ice flow velocity increase. The past decade has seen extreme Antarctic-wide sea-ice minima in 2017, 2022, 2023 and 2024, and a sea-ice decline in 2014–2016. There is a growing concern that there is a structural change taking place in the Antarctic atmosphere-ocean-sea-ice system (Purich and Dodderidge, 2023; Hobbs et al., 2024), revealed by analysis of reconstructed and satellite-observed sea-ice extent that shows increasing persistence in sea-ice extent anomalies, a pattern especially clear in East Antarctica (Raphael et al., 2025).





### 4.4 Future unpinning at Holmes West Glacier

The current grounding line position at Holmes West Glacier suggests that this glacier may be vulnerable to MISI in the near future. Our findings show the grounding line was pinned on a local bedrock high between 1996 and 2014, which possibly inhibited mCDW from accessing the grounding line (Fig. 9d). The 2014 grounding line position is at the very edge of a steep

retrograde bed and, over the next 2 km inland along the flowline, the bed elevation drops by 650–700 m (Morlighem, 2022, Pritchard et al., 2024). However, we note that the reverse-sloping bed topography is subject to some uncertainty. The BedMachine3 bathymetry suggests the bed is reverse sloping for 3–4 km then stabilises; though, this bathymetry was produced using mass conservation in the absence of radar data (Morlighem et al., 2020). Comparatively, the Bedmap3 reverse slope continues inland for the next 25 km along the Holmes West Glacier flowline; this was extracted from numerous radar flight

paths that intersect the trough, albeit sparsely spaced compared to Totten or Moscow University troughs (Pritchard et al., 2025). Regardless, given its current thinning rate of −1.22 m yr$^{-1}$ at the grounding line, Holmes West Glacier will likely unground from the local bedrock high and retreat to become grounded on a bed below 1000 m depth, where mCDW could reach the grounded ice, exposing it to rapid melting. Given the enhanced ice-shelf basal melting (Adusumilli et al., 2019), thinning across the ice shelf and grounded ice, continued grounding line retreat, and observations suggesting mCDW at depth, future

grounding line retreat could theoretically be sufficient to initiate MISI at Holmes West Glacier. This is very concerning given that the Holmes catchment stores 11 cm of SLE (Rignot et al., 2019). These findings suggest it is a priority to model this region, especially Holmes West, and collect more oceanographic and bathymetric data to understand the timescale of future sea level contributions.

     Furthermore, there is an adjacent 4700 km$^2$ ice rise directly west along the coastline from Holmes West Glacier

(Matsuoka et al., 2015) that may exert a buttressing force on the ice sheet, retarding the flow of grounded ice to the ocean and contributing to grounding line stability (Favier and Pattyn, 2015; Matsuoka et al., 2015). If the current rates of grounding line retreat (−70 m yr$^{-1}$) continue, Holmes West Glacier could unpin from the bedrock high in the next decade, losing contact with the stabilising ice rise, potentially resulting in threshold-like behaviour (Matsuoka et al., 2015).

     Although we observed greater grounding line retreat at Holmes East and Frost glaciers, the likelihood of irreversible

grounding line retreat or MISI is less concerning given the elevation of the glacier troughs along the flowlines. The bed inland of the Frost Glacier grounding line is relatively flat and the bed is prograding inland of Holmes East Glacier grounding line (Fig. 9). It is generally understood that retrograde slopes favour more extensive grounding line retreat for a given basal melt rate (Milillo et al., 2019; Millan et al., 2022).





## 5 Conclusion

To investigate recent glacier dynamics in Porpoise Bay, Wilkes Land, East Antarctica, we measured changes in ice-shelf calving position, ice surface velocity, grounded ice surface elevation, and grounding line position of four outlet glaciers over the last three decades. The observed glacier dynamics were compared to sea-ice conditions and ocean temperatures and salinity to assess the potential forcing of any observed changes. Despite large uncertainties, our results suggest Frost and Holmes East glaciers have undergone substantial grounding line retreat, with more modest rates of grounding line retreat observed elsewhere

in Porpoise Bay. Observations of surface elevation change depict considerable thinning at Frost, Holmes East, and Holmes West glaciers between 2003 and 2019, which propagated inland, with Holmes West Glacier displaying potentially one of the highest rates of thinning in East Antarctica. Furthermore, the outlet glaciers underwent moderate increases in velocity over the study period. Taken together, these findings are consistent with dynamic changes in this region, consistent with previous work suggesting that it is losing mass and contributing to sea level rise (Rignot et al., 2019; Adusumilli et al., 2020; Smith et al.,

2020a; Rignot et al., 2022). In addition, we report on a new, previously unidentified calving event in 2021/22, alongside near-synchronous ice-shelf calving events in 2007/08 and 2016/17. As in previous work (Miles et al., 2017), we show that the calving events were caused by a reduction in sea ice in the preceding winter and summer and the collapse of a supportive mélange.

Our findings of dynamic change are consistent with observations of mCDW at the shelf break seaward of Porpoise
Bay, with EN4 ocean temperature data suggesting the presence of mCDW at depths below 450 m on the continental shelf. We find that bathymetric pathways beneath Frost, Holmes East and Holmes West glaciers could transport mCDW to the sub-ice cavities, driving high rates of basal melting that have been reported in previous work (Adusumilli et al., 2020). Furthermore, we find Frost and Holmes East glaciers are grounded sufficiently deep for mCDW to directly access the grounding line. In contrast, Holmes West is grounded on a local bedrock high that could inhibit year-round access of mCDW to the grounded

ice, which is also consistent with its relatively low grounding line retreat rate. Of concern is that Holmes West Glacier is currently grounded at the edge of a steep retrograde bed that would see the grounding line lose up to 700 m elevation in less than 2 km retreat along the flowline, providing access for mCDW to melt the grounded ice, which could initiate rapid and irreversible grounding line retreat down a retrograde bed that continues on a reverse slope for 25 km inland. If MISI is initiated, the retreat could rapidly drain Holmes drainage basin, contributing 11 cm to global sea level.




*Data availability.* Ice shelf calving front positions, central flowlines, sampling boxes, and grounding lines positions are available to download from Zenodo (https://doi.org/10.5281/zenodo.15847590; Weatherley et al., 2025). The monthly ice surface elevation change dataset from Schröder et al. (2019b) is available from PANGEA
(https://doi.pangaea.de/10.1594/PANGAEA.897390) (1978–2017), from Nilsson et al. (2023) is available from NSIDC (https://doi.org/10.5067/L3LSVDZS15ZV) (1985–2020), and from Smith et al. (2020b) is available from the Research Works Archive (https://doi.org/10.5067/L3LSVDZS15ZV). The 2000–2022 ITS_LIVE annual velocity mosaics (Gardner et al., 2022) are available from NSIDC (https://nsidc.org/data/nsidc-0776/versions/1). The MEaSUREs 1996 grounding line positions (Rignot et al., 2016) are available from NSIDC (https://doi.org/10.5067/IKBWW4RYHF1Q). The ASAID 2001
grounding line positions (Bindschadler and Choi, 2011) are available from the US Antarctic Program Centre (https://doi.org/10.7265/N56T0JK2). The MOA grounding line positions from 2004 (Haran et al., 2021a), 2009 (Haran et al., 2021b), and 2014 (Haran et al., 2018) are available from NSIDC (https://doi.org/10.5067/68TBT0CGJSOJ), (https://doi.org/10.5067/4ZL43A4619AF), and (https://doi.org/10.5067/RNF17BP824UM). The 1979–2025 monthly sea ice concentration dataset (Fetterer et al., 2017) is available from NSIDC (https://doi.org/10.7265/N5K072F8). The MEOP data
(Treasure et al., 2017) is available from https://meop.net/database/. The EN4 subsurface ocean temperature objective analyses (Good et al., 2013) is available from the Met Office Hadley Centre (https://www.metoffice.gov.uk/hadobs/en4/). The Bedmap3 bed elevation, ice surface elevation, and ice thickness datasets (Pritchard et al., 2024) are available from NERC EDS UK Polar Data Centre (https://doi.org/10.5285/2d0e4791-8e20-46a3-80e4-f5f6716025d2). The BedMachine3 bed elevation (Morlighem, 2022) is available from NSIDC (https://doi.org/10.5067/FPSU0V1MWUB6). The ANTGG bed elevation (Rignot
et al., 2024) is available from Dryad (https://doi.org/10.5061/dryad.rbnzs7hkc). The REMA elevation dataset (Howat et al., 2022) is available from Harvard Dataverse (https://doi.org/10.7910/dvn/x7ndny).

*Author contributions.* MW, CRS, and SSRJ contributed to the design of the initial project. MW undertook the data collection and led the analysis, with guidance from CRS and SSRJ. SR completed the DInSAR grounding line mapping within Porpoise
Bay, with MW writing up the method, results, and figures, with additional figure contributions from AS. MW led the manuscript writing, with all authors editing the manuscript.

*Competing interests.* The authors declare that they have no conflict of interest.

*Acknowledgements.* SSRJ is supported by Natural Environment Research Council (NERC) UK research grant NE/Y00115X/1. We are grateful to Hannah Picton and Bertie Miles for their guidance.



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
