# Peer review of "Dynamic Thinning and Grounding Line Retreat in Porpoise Bay, Wilkes Land, East Antarctica"

_EGUsphere, 2025_

## Referee Comment (RC3)

Comments on "Dynamic Thinning and Grounding Line Retreat in Porpoise Bay, Wilkes Land, East Antarctica"
by Weatherley et al. 2025

This paper evaluates dynamical changes of the Porpoise Bay glaciers in East Antarctica from 1963-present. They present findings on surface elevation changes, grounding line migration, velocity changes, and calving events. They also include analysis on sea ice concentration and oceanographic data as potential drivers of such changes.

The authors conclude that the glaciers are indeed undergoing dynamical changes. They attribute elevation changes and grounding line migration primarily through the presence of mCDW that is eroding the ice shelf base at the grounding line. They attribute ice front and velocity changes primarily through changing sea ice concentrations, which are suggested to provide buttressing to the ice fronts.

The paper is well written with pleasant-to-look-at and easy-to-understand figures. The authors used an impressive variety of datasets to evaluate the changes and potential forcings in this region. They fully attempt to synthesize all of this information together in a readable, understandable format. However, there are several aspects of the paper that need revision before publication.

**Main Comments**

Grounding line positions – I note that the authors do a good job qualifying their results with caution due to the inherent uncertainties in the different grounding line positions. However, given the expressed uncertainty in the datasets and DInSAR analysis (and the bedrock data) the strong conclusion that MISI is inevitable in the near future and the region will certainly contribute 11 cm of SLR seems a bit bold. I suggest the authors consider two changes to support their conclusions. 1) regarding the grounding lines – I understand the difficulties with the DInSAR coherence, have you considered mapping a more recent grounding line manually with the optical imagery or elevation data that you already include in the manuscript? It would strengthen the conclusions to see a GL change using a consistent grounding line proxy method. Especially because the hinge line (from SAR) is landward of the true grounding line, which could be skewing your results. 2) you mention that there is 47,000km+ of airborne radar data and yet none of it can contribute to your understanding of the bedrock. Given the uncertainties in Bedmap3 and Bedmachine and that the seal tag data shows bathymetry 500m deeper than what was reported in the bathymetry data, I advise revisiting the radar data to support your conclusions on the possibility of MISI. Perhaps there is a radargram you can include in supplemental information that shows some form of a retrograde slope or pinning point.

Dynamical changes in relationship to sea ice and calving events – the authors suggest that there is a significant link of the ice flow velocities to sea ice and mélange presence and ice front calving. However, this claim either needs more detailed analysis, or as reviewer 2 points out,

could be removed from the study altogether. For a more detailed analysis, the authors could provide the precise timing of the calving events (days/months, at least for the later years when the imagery is better) and the conditions of the sea ice and mélange within that same time period. As sea ice and mélange can vary week to week (and sometimes even day to day) an annual assessment is not robust enough for this conclusion, especially given that the extent of the relationship between sea ice and ice shelf calving events is not yet fully established within our current level of understanding. It's also not clear how the flow velocity of the glaciers and ice shelves were affected by mélange presence, and including more evidence is paramount for establishing this dynamical link.

Dynamic thinning – I think this requires a bit more strengthening in the discussion. I suggest adding more discussion of velocity changes in other glaciers in the region as reference for what quantitatively constitutes "speed up". For example, you report velocity increases ranging from 2-18% but how much of that is natural variability in ice flow speed and how much of that could be caused in changes in the upstream ice flux? There is also some weird stuff happening at HW where the inland ice is slowing down before the GL ice and that is not discussed at all. You present clear evidence of thinning, but the relationship to velocity changes needs to be fleshed out more. This can be done by putting the % changes in context to other known glacier accelerations as well as including some discussion on upstream surface mass balance changes that could contribute to changes in ice fluxes and subsequent velocities. If you want, you could even provide a strain rate map to show this (perhaps this turning into a second paper :P).

Positioning of floating tongue velocity box on Frost Glacier in Figure 1 – it appears that it is in grounded ice relative to the 96' GL position. Given the uncertainties in the GL migration, I suggest moving the box to floating ice, which is floating for all GL positions.

Please export figures as 300-400 DPI to ensure high resolution for all of them.

**Minor Comments**

Line 26: References for "recent work has mostly…" it would be nice to include some newer references as some of these are becoming outdated by now!

Line 29: when mentioning the SLR amounts, include the AP as well (perhaps in the sentence above).

Line 34: the way this sentence is written it is unclear if the EAIS has larger uncertainties compared to WAIS and AP or if it simply has "large uncertainties", please correct.

Line 54: I'd be careful when saying a "recent" study as "recent" is subjective.

Line 58: Does Bedmap3 show the same thing? I'd reference Pritchard here then, too.

Figure 1: is there a reason to show Bedmap3 vs. Bedmachine as the background image?

Line 94: The way this reads it sounds like you're looking at all those attributes from the 60's to present day, but that's a bit misleading given it's only the ice fronts that go that far back.

Line 117: This sentence is a bit confusing, perhaps to clarify use "as clouds conceal…."

Line 121: Here and throughout, there is a lot of talk of uncertainty being 1 pixel but that is meaningless unless we know the resolution. Can you change it throughout so that it is a more quantitative? I'd also like to see some mention of the temporal gaps in the data from the early years and how that could affect the results.

Line 127: see above.

Line 135: How does your 10 km$^2$ boxes fit with the two datasets grids? Does your box span multiple grid cells or is it the same? If the former, how did you account for that? Also, if you extracted the elevation change at the GL, which GL is this referring to and how could your results change if the GL moves?

Line 138: You begin this section by saying that the monthly elevation changes are from 1985 but now you say you're using them from 1992 onwards, please be clear about what time period you are using and how you calculated them.

Line 144: grounding finding errors – what is that?

Line 144 part 2: I'd move this last sentence to the paragraph above because it is confusing to jump from Smith back to Nilsson and Schröder.

Line 159: pixel scale again (see above)

Line 165: Can you more clearly explain what velocities were thrown out and what was kept?

Line 183: How was the 2020 buffer zone determined? Why is 10km your maximum uncertainty?

Line 198: Add in the parenthesis that hinge line is landward of true grounding line.

Line 209: Please see main comment about radar data.

Line 215: If AntGG2021 is constrained by Bedmachine3, and Bedmachine3 has uncertain bed topography, how does that propagate into the AntGG2021 dataset?

Line 220-225: How do the differing time periods of REMA, Bedmap3, and Bedmachine influence the results of the GL migration?

Line 234: maybe specify here what denotes "surface" "intermediate" and "deep" depths, and provide a reference which depths are commonly used for this region (the depths mentioned in Figure 11)

Line 234 part 2: can you elaborate on this? "*We acknowledge that the nature of EN4 analysis data creates very high uncertainty estimates…*"

Line 242: If the data are not spatially or temporally continuous how did you process them such that they ended up in Figure 12 (speaking of – what is that nearly straight line of the continental shelf going NW?)

Line 250: Do you mean high cloud liquid *content*?

Line 255: Relative to the upstream point in the box method, right? Can you add that reference line in figure 2?

Line 266: I suggest switching the sentence so that it says there were several calving events and advances but no clear trend over the study period.

Line 267: Can you provide the percent area change? I think it's important to be able to compare the %area change, especially because you reference Fürst in the discussion and his 18.3% threshold, for example if 8km of retreat for Glacier 1 is 30% of its ice shelf area that makes a much more convincing argument for changes in backstress to the upstream grounded glacier, but if 8km of retreat is only 0.05% then that seems pretty negligible.

Line 270: Is it possible to find a more precise timing of the calving events of the different glaciers? (See main comment)

Section 3.1 – you might consider putting these in a table and only discuss the few major events in the text.

Figure 2: On Panel A, I suggest changing the smaller panel ordering so that they read from left to right. For the Landsat image, can you provide the date and whether or was Landsat 8 or 9. It's also not clear why different ice shelf extents are shown for different dates at the different glaciers, can you explain the date choices in the text and the reasoning? Why is it starting in 1973 instead of 1963 for the close ups? This seems like an odd choice to me.

Figure 3: Why present it relative to the 1963 line when you did the box method?

Line 307-308: Looking at Figure 4, this is not evident – can you provide perhaps some statistics to back up this claim? Is there a statistically significant speed change before or after the calving events?

Figure 4: I'd make your definition of major and minor calving events consistent across your figures, the definition here is different than that mentioned in Figure 13. Also, why does HW speed up inland but not at the GL or FT In 2020?

Line 318: what is the "ca" here?

Line 324: Are you missing a – sign for 0.003?

Line 331: Perhaps my perspective from West Antarctica is skewing this interpretation, but why do you call thinning of <0.5m per year "rapid"? Is that considered "rapid" for this region? For context, some glaciers have thinning greater than 40 m/yr, but this is in Patagonia, the AP, etc.

Line 335: Is there not any data from Schröder for Holmes West? If not, why not? Please mention this in the text/methods.

Figure 7: See comment about Frost "floating tongue" being grounded. Also, perhaps here you can show the different grounding lines? Also, is that the black dashed line?

Figure 9: In the figure caption, it would be nice to be reminded of what the methods of the datasets are. (1996 SAR, 2014 optical break in slope, etc.). Also, make sure you check the font sizes of the years because it looks like there is a difference in Panel C for "1996".

Line 403: Did you calculate a linear regression for the temperature (if so, perhaps display it on Figure 11)? And by significant, do you mean statistically significant?

Line 413: I think it's really cool you used the MEOP casts. Given the temporal and spatial resolution, can you be clearer about what the data provide for your analysis? Is it just characterizing the ocean near the bay during the 17-year period of the dataset? Since you only provide the averages (I think? See comment in methods, line 242) it makes me wonder if it there are changes occurring or not. Is it possible to parse out that information (e.g. has there always been mCDW? In your discussion you say that there is an "intrusion" of mCDW, but I wonder if there is no warming trend in the EN4 data and you don't present a trend in the MEOP data, what does that say about the GL migration?).

Figure 11: Can you show some of the MEOP casts on this plot?

Figure 12: It would help the reader if panel. C was labelled "depth of observed maximum temperature" or at least included in the caption. What year is the MEaSURES ice shelf edge from?

Line 437: How do you explain the significantly low sea ice concentration years that don't have any major calving events?

Figure 13: The pale gray boxes are covering the x-axis tick marks, please fix this. Also, can you make them thinner to match the timing of the calving events? (See previous comment about timing of calving events).

Figure 14: Is it possible to perhaps combine this figure with Figure 13? Or put one of the figures in the supplemental information. It is not clear what new information this is telling us. Also, are the asterisks showing when any of the four glaciers calve or only when they all do?

Line 449: albeit, the magnitude is quite different in W. Antarctica. (For example, Thwaites lowering at the GL is 3-7 m/yr! Wild et al., 2021.)

Table 3: Please also include the bed elevation change for Bedmachine and AntGG2021. How were "strong, moderate" and "minimal/inconsistent" chosen? Please include more details on this decision making in the text. For d) How did you calculate this? Displaying this information as m/yr suggests it is retreating, but you provide evidence for it advancing as well. Instead, can you show simply the difference (in m) in ice front edge in 2025 compared to 1963?

Line 491-499: See main comment about dynamic thinning.

Line 502: do you mean km?

Line 504: Interesting, can you explain how this relationship works when the glaciers are flowing less than 800 m/yr?

Line 505: Looking at Figure 4, it's not clear that there is significant speed-up occurring, except maybe at Holmes East, can you elaborate on this statement then?

Line 508: See my comment about the MEOP casts and an analysis through time.

Line 513: Where in East Antarctica were these measurements taken? Were they near your study site?

Line 544: Which bathymetry is being referred to here and how does that effect your interpretation of the bedrock data and grounding line retreat?

Line 578-580: It would be great to show the images of the break-up and removal of the mélange and sea ice as a supplemental figure.

Line 582: How near-synchronous? When did the calving events occur relative to the removal of sea ice and mélange?

Line 592-610: This is where I think it would be useful to include the %area change of the calving events as I mentioned above.

Line 598-600: What about the sea surface slope suggested by the previous reviewer? I'm not sure there is enough evidence here to claim such a strong link.

Line 609: If the sea ice was exerting a backstress on the glaciers and slowing them down, why would you see a greater decrease in speed in the inland ice than the grounded ice for HW? This is opposite of what you would expect.

Line 615: This claim would benefit from including a citation of sea ice slowing down ice shelves.

Section 4.4: This section would improve significantly if there was any kind of radargram available to confirm the bedrock datasets.

Line 633: 2019 should be 2020

Line 657: See previous comment about velocity increases.

Line 664: See comment about supplemental figure showing support mélange collapse.

Line 673: How long is the reverse slope in all the different bedrock datasets? Is 25k the longest? What is the shortest?

Line 674: I think this is quite the bold statement given the uncertainties in the bedrock and the grounding line migration. I'd suggest toning it down a notch!